# Managing New Risks of and Opportunities for the Agricultural Development of West-African Floodplains: Hydroclimatic Conditions and Implications for Rice Production

**Aymar Yaovi Bossa** [1,2,*], **Jean Hounkpè** [1,2], **Yacouba Yira** [1,3], **Georges Serpantié** [4], **Bruno Lidon** [5], **Jean Louis Fusillier** [5], **Luc Olivier Sintondji** [2], **Jérôme Ebagnerin Tondoh** [6] and **Bernd Diekkrüger** [7]

[1] West African Science Service Centre on Climate Change and Adapted Land Use (WASCAL), Ouagadougou, Burkina Faso; hounkpe.j@wascal.org (J.H.); yira.y@wascal.org (Y.Y.)

[2] National Water Institute, University of Abomey Calavi, Cotonou, Benin; o_sintondji@yahoo.fr

[3] Applied Science and Technology Research Institute–IRSAT/CNRST, P.O. Box 7047, Ouagadougou, Burkina Faso

[4] Institute for Research and Development—IRD-UMR GRED-UPV, 34090 Montpellier, France; georges.serpantie@ird.fr

[5] Centre for International Cooperation in Agronomic Research for Development—CIRAD-UMR G-eau, 34090 Montpellier, France; bruno.lidon@cirad.fr (B.L.); jean-louis.fusillier@cirad.fr (J.L.F.)

[6] UFR des Sciences de la Nature, Université Nangui Abrogoua, 02 BP 801 Abidjan 02, Cote D'Ivoire; jetondoh@gmail.com

[7] Department of Geography, University of Bonn, Meckenheimer Allee 166, 53115 Bonn, Germany; b.diekkrueger@uni-bonn.de

\* Correspondence: bossa.a@wascal.org

**Abstract:** High rainfall events and flash flooding are becoming more frequent, leading to severe damage to crop production and water infrastructure in Burkina Faso, Western Africa. Special attention must therefore be given to the design of water control structures to ensure their flexibility and sustainability in discharging floods, while avoiding overdrainage during dry spells. This study assesses the hydroclimatic risks and implications of floodplain climate-smart rice production in southwestern Burkina Faso in order to make informed decisions regarding floodplain development. Statistical methods (Mann-Kendall test, Sen's slope estimator, and frequency analysis) combined with rainfall—runoff modeling (HBV model) were used to analyze the hydroclimatic conditions of the study area. Moreover, the spatial and temporal water availability for crop growth was assessed for an innovative and participatory water management technique. From 1970 to 2013, an increasing delay in the onset of the rainy season (with a decreasing pre-humid season duration) occurred, causing difficulties in predicting the onset due to the high temporal variability of rainfall in the studied region. As a result, a warming trend was observed for the past 40 years, raising questions about its negative impact on very intensive rice cultivation packages. Farmers have both positive and negative consensual perceptions of climatic hazards. The analysis of the hydrological condition of the basin through the successfully calibrated and validated hydrological HBV model indicated no significant increase in water discharge. The sowing of rice from the 10th to 30th June has been identified as optimal in order to benefit from higher surface water flows, which can be used to irrigate and meet crop water requirements during the critical flowering and grain filling phases of rice growth. Furthermore, the installation of cofferdams to increase water levels would be potentially beneficial, subject to them not hindering channel drainage during peak flow.

**Keywords:** inland valley development; hydroclimatic hazard; water control structure; sustainable rice production

## 1. Introduction

The future of West Africa, and its economic, political, and social balance, depend on the ability of the agricultural sector to adapt and ensure food security under multiple pressures, such as climate change and demographic growth. In Africa, only 12.5 million hectares are irrigated out of a total of 202 million hectares of cultivated land, or 6.2%. The proportion of irrigated land in the south of Saharan Africa is even smaller, with only 5.2 million hectares, or 3.3%, of cultivated land being irrigated [1]. The increase in population will have serious implications in terms of agricultural production and the availability of natural resources. Adaptive strategies to help cope with the potential decrease in crop yields include promoting the extensive development of inland valleys in West Africa. This is because of their great potential as rice-based production systems due to the high and secure water availability and soil fertility [2]. As such, the West African floodplains are privileged places for agricultural intensification, but play a diminishing role in the face of droughts that affect rainfed crops. Key factors of concern for the agricultural development of floodplains include flood hazards, surface flow deficits due to dry spells, and early flood recession. The valorization of floodplains faces numerous technical, social, and economic constraints that involve an intensification of crops and hence new risks linked to climate change. These are characterized by increased irregularities in rainfall, onsets of extreme floods, and long-lasting droughts.

As a landlocked country, Burkina Faso is vulnerable to climate variability [3]. This variability not only occurs at a daily, seasonal, and interannual scale, but can also be multidecadal. A break in annual rainfall was observed during the 1970s, irrespective of latitude or longitude, in West Africa [4]. It was especially prominent in the savanna, which includes the study area of Dano. The causes of this prolonged drought, subcontinental in scale, remain controversial and undoubtedly multifactorial and multiscalar. Many authors have shown, in addition to the global natural variations (i.e., astronomical, oceanic, and volcanic), the anthropogenic effects at different scales. These are observed at a regional (increase in the albedo effect because of the rapid urbanization of the Sahel, and deforestation of the lower coast reducing real evapotranspiration (ETR) and increasing flow), intercontinental (European air pollution of the 1970s, favoring regional cooling, i.e., anticyclonic conditions in regulatory pathways), and global (greenhouse gas-related climate change and its effects on heat and excessive events) level [5]. A change in hydroclimatic conditions can substantially modify the hydrological regime of an inland valley and its drainage area [6]. This modification can result in flooding or drying conditions in the lowlands, implying a possible reduction in its productivity. Furthermore, land cover and land use change can alter the floodplain and impact its ecosystem [7], but investigating this is beyond the scope of this study. Given the uncertainties in climate model predictions (especially for precipitation), analyzing the current climate hazards using observed data and their possible implications for the future is required.

Burkina Faso's agricultural sector continues to generate approximately one-third of the country's GDP and employs 80% of the population [8], despite the harsh climatic condition. Notwithstanding the importance of agriculture in the economy of Burkina Faso, the sector is facing many challenges, including threats from many natural disasters, such as floods, droughts, and violent winds, which lead to low crop and livestock productivity [9]. Since 1970, investment has been made by the government of Burkina Faso to address the issue posed by hydroclimatic risks. This includes developing rice production intensification policies in inland valleys that encompass physical development, the social organization of production, material support, organization of the rice sector, subsidies, and legal connotations. Subsequently, 10% of the inland valleys suitable for agriculture have been developed in southwestern Burkina Faso. However, as reported by the regional agriculture extension service, up to

30% of the developed inland valleys have been abandoned due to increasing hydroclimatic hazards. There is therefore a need to describe the seasonal, average, and frequency characteristics of the climate that can impact rice production in the region.

The objective of this work is to analyze the hydroclimatic hazards by considering the period of 1922–2017 and their implications for rice production in southwest Burkina Faso to support agricultural policies for adequate water infrastructural development. Two research questions are considered: (i) What are the current trends in climatic and hydrological hazards and what are their implications for food production in inland valleys? (ii) What are farmers' perceptions of the hydroclimatic risks in the region, and what strategies have consequently been developed to face the challenges encountered?

## 2. Materials and Methods

This section is divided into five sub-sections, which are the study area and data used, the various modalities of lowland development (traditional vs. modern development), climate-related local knowledge and hydroclimatic variables' analysis, rainfall–runoff modeling and frequency analysis, and water availability evaluation during the critical phase of rice development.

### 2.1. Study Area and Data Used

The case study areas are the Lofin catchment and Lofin inland valley, located in the municipality of Dano in the southwest region (région du Sud-Ouest) of Burkina Faso, West Africa (Figure 1). Dano is situated in a tropical climate region with a unimodal rainfall regime (Figure 2). The mean annual rainfall is approximately 921 mm, with a standard deviation of 106 mm for the period 1980–2018. The annual rainfall regime is characterized by the alternation of two contrasting seasons: a dry season from November to March, in which rainfall is almost absent (58 mm in October, the driest month), and a rainy season from April to October (average 238 mm in August, the wettest month) [10]. The climatic water demand (ET0) is more stable during the year, but it varies, on average, from 123 mm in August to 175 mm in April [10]. Long-cycle crops (120 days), such as rice sown after the 10th of July, are at risk of water stress in the middle and end of the growing cycle. The average annual temperatures between 1970 and 2013 ranged from 25 to 33 °C and from 25 to 31 °C at the Boromo and Gaoua stations, respectively (Figure 2). The annual insolation varies between 6 and 8 h/day, and the air humidity ranges from 35% to 80%. The dominant vegetation comprises shrubs and/or the tree savanna type, and resulting successional vegetation from the degradation of cleared forests [11]. This is due to both human activities and the dry period since 1970.

In this region, wetlands have historically been used as pasture in the dry season. Rice was one of the first crops cultivated in these areas (Figure 3). From the 20th century onwards, the agricultural use of the inland valleys, referred to locally as 'bas-fonds', was fostered because of population growth and migratory flows. Currently, in addition to rice, wetland use has been diversified with other crop types, such as vegetables, fruits, and cereals [12]. Rice products are mainly intended for consumption, with an increasing share of rice in the local food.

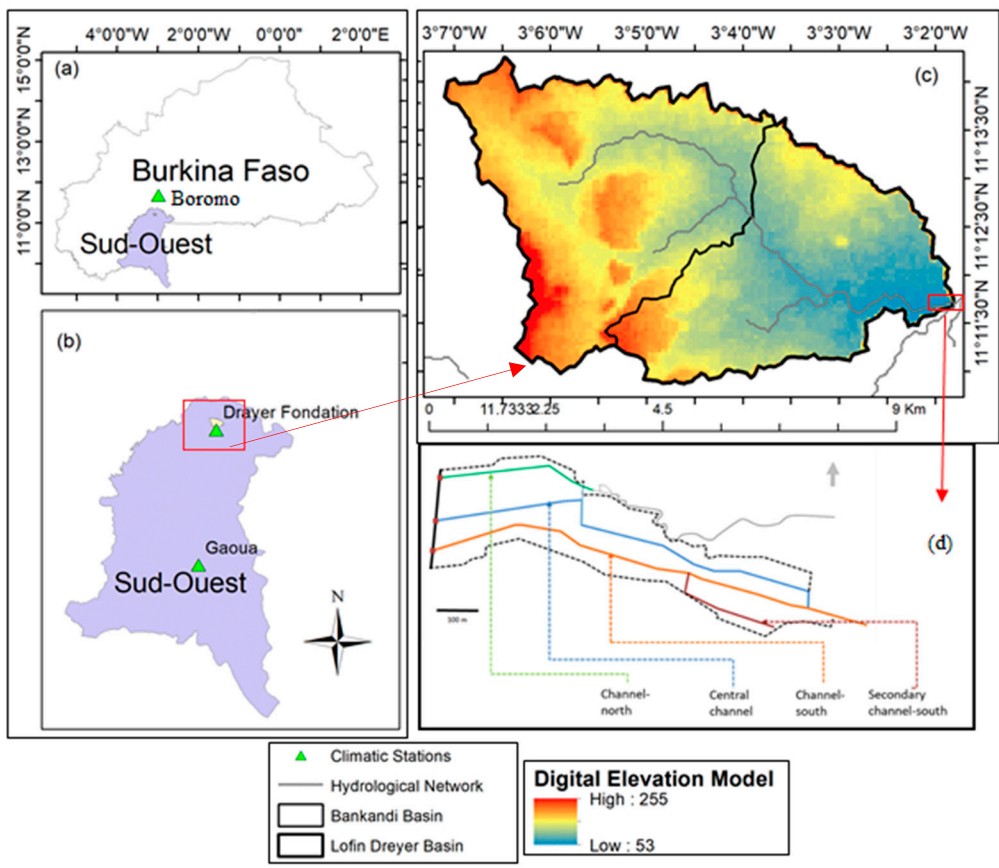

**Figure 1.** (**a**) Location of the study area in Burkina Faso. (**b**) The southwest region. (**c**) The Lofin catchment. (**d**) The Lofin inland-valley with irrigation/drainage channels.

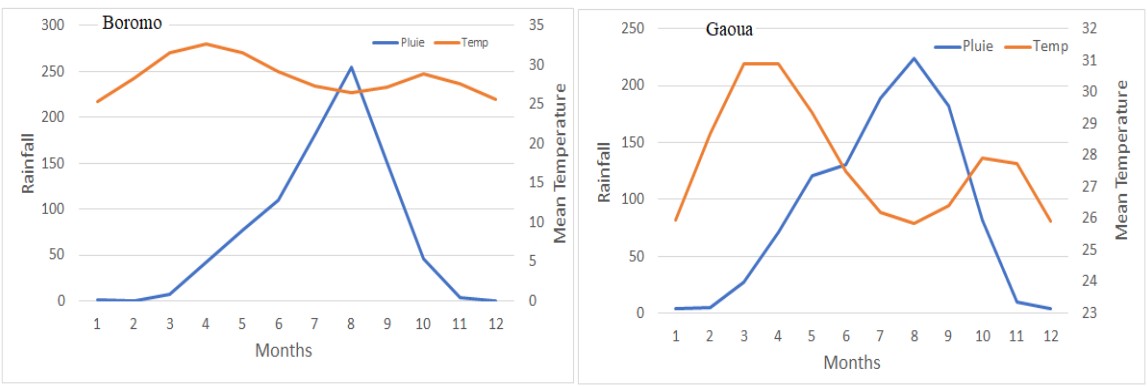

**Figure 2.** Rainfall and temperature at Boromo and Gaoua climatic stations.

The data used in this study are rainfall data of the Boromo and Gaoua stations from 1922 to 2016, and rainfall data of the Dreyer Foundation from 2017 to 2018. Discharge data of the Lofing-Radier station from 2017 to 2018 were measured by WASCAL (www.wascal.org) during project implementation. Other climate data, such as the minimum and maximum temperature, sunshine duration, wind speed, and minimum and maximum relative humidity from 1922 to 2016 of the Boromo and Gaoua stations were used and obtained from the Burkina Faso national meteorological directorate (including long-term rainfall data).

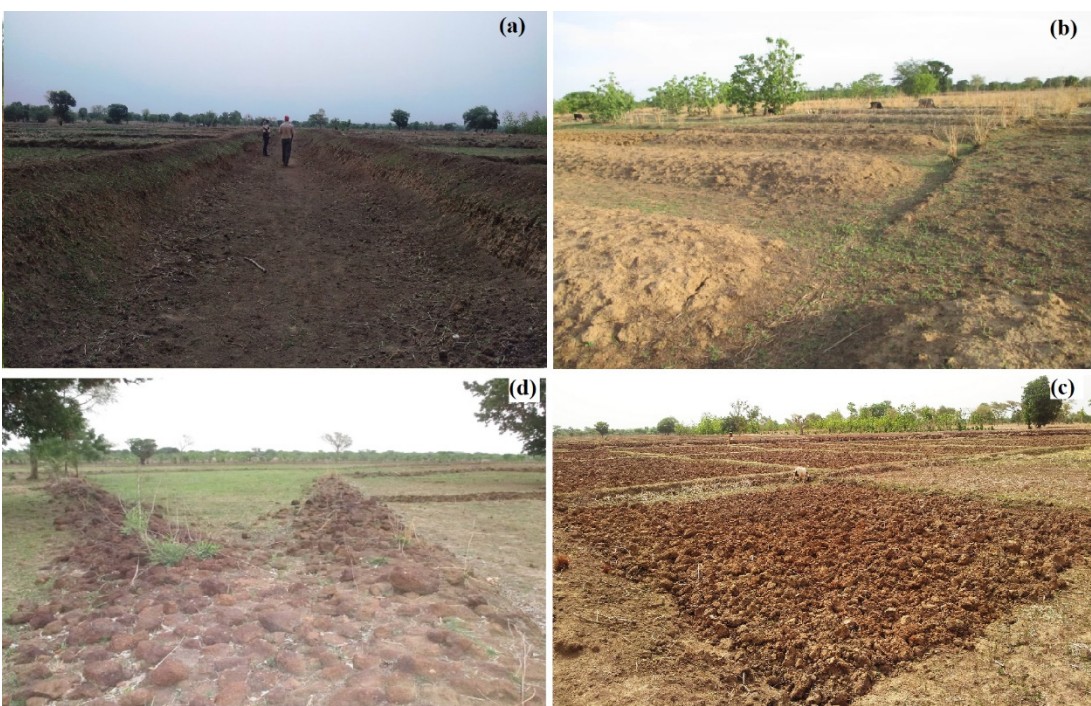

**Figure 3.** Different types of lowland development, including a rice field model with sprinkler drains (**a**,**b**), cyclopean concrete pouring dikes with a central cofferdam (**c**), and compacted clay bunds protected by geotextiles and rocks following the level curves with openings (**d**).

## 2.2. Various Modalities of Lowland Development: Traditional vs. Modern Development

Four types of lowland development have been observed and can be classified into two main groups: traditional and modern lowland designs. Traditional lowland development is a combination of the techniques developed by farmers for managing the agrarian space, the water, and the various types of production. The objective pursued by the farmers is to grow crops while minimizing the risks associated with drought and flooding. Schemes designed by farmers for the control and management of water in the lowlands include, firstly, large ridges arranged perpendicularly to the water flow direction, and secondly, large ridges in the shape of a contour dike with gaps.

Baffles are formed that not only slow the flow of water favoring infiltration, but that also enable the management of a water level in the grooves between the ridges. Upland crops (maize or tobacco) are placed on the ridges. Sorghum and taro are placed on the flank of the ridges. Rice, a water-demanding crop, is sown and transplanted into the furrows. This polyculture system is adapted to local conditions, has a low associated cost, and is more resilient to the risk of climatic disasters.

In contrast, modern lowland developments are those designed and implanted by external organisations, such as funded projects and programs, and NGOs. The principle objectives of such developments are multi-functional, aiming to

- Partially control water through the installation of hydraulic structures;
- Distribute the water at the landscaped site;
- Optimize the drainage of flood waters;
- Avoid and minimize the adverse effects of water shortages due to dry spells during the crop season; and
- Support non-seasonal crops, if possible.

To achieve these objectives, several types of development were designed, of which three (3) types of models are described. First is the rice field model with sprinkler drains (Figure 3). This model consists of channeling runoff by following preferential paths marked by the differentiation of surface elevation.

The canals are used for irrigation and drainage. The individual plots are partitioned by small bunds which the producers can open to irrigate their plants. This model is promoted in the area by the Dreyer Foundation. Second are compacted clay bunds protected by geotextiles and rocks that follow the level curves with openings. This model is a flood spreading arrangement, with the possibility of drainage being provided by the openings. The third model is the cyclopean concrete pouring dikes with a central cofferdam. This model is based on the threshold for slowing the flow of water on the course bed, which leads to a substantial change in the height of the water level, and is then managed using the cofferdam.

With these three models of landscape control, adding garden plants arranged with wells is necessary. The modern development models of lowlands strongly alter the hydrology of the valley bottom. Although this may be advantageous, it also adds new constraints that can become risks, depending on the physical and social environment.

### 2.3. Climate-Related Local Knowledge and Hydroclimatic Variables' Analysis

A survey of farmers' perceptions on climate and climatic changes was conducted in the Lofing lowland in 2017 using a questionnaire. A total of 17 farmers were randomly selected, and a questionnaire was administered individually. The hydroclimatic variables were statistically analyzed using the quantile method, Mann-Kendall test, and Sen's slope estimator. Different time steps were considered to aggregate the time series over 10-day, monthly, and annual time scales. Reference evapotranspiration was computed using the Food and Agriculture Organisation (FAO) ETo calculator software based on the Penman–Monteith formula [13]. The rainfall onset and cessation dates were defined using the Franquin (1969) [14] method. At the 10-day scale, the rainfall onset corresponds to the date from which rainfall is greater than half of the potential evapotranspiration (R > ET0/2). The methodological framework of the study is presented in Figure 4. It shows the different steps fulfilled to perform this study. The Mann-Kendall test [15,16] is a nonparametric trend detection method widely applied to hydroclimatic variables [17–19]. The null hypothesis $H_0$ for the test is that there is no trend in the time series, while for the alternative hypothesis $H_1$, there is a significant trend in the time series at the 0.05 significance level. This is a robust test in the sense that it does not make any assumptions about the distribution of variables. In addition, the Sen slope method [20] is considered for estimating the magnitude of the slope if a trend is detected in the time series.

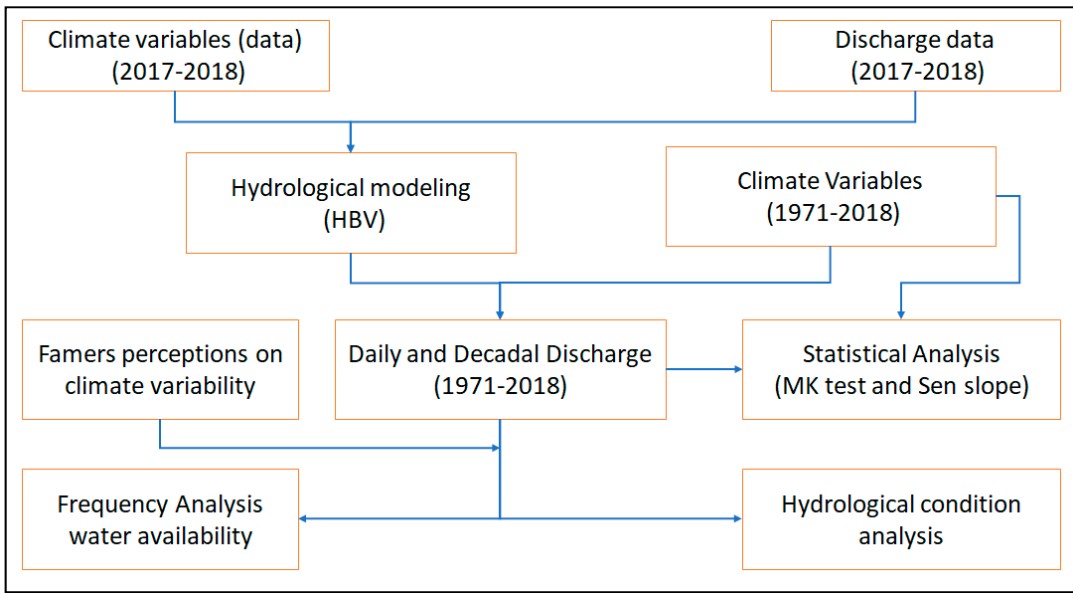

**Figure 4.** Methodological framework of the study.

### 2.4. Rainfall–Runoff Modeling and Frequency Analysis

To further access the hydrological aspect of the study area, an HBV model [21] was calibrated and validated for the Lofing basin at the outlet of Lofing Radier. HBV is a conceptual, lumped, and time-continuous hydrological model that simulates discharge using rainfall and potential evaporation as inputs. The HBV model has four main component routines: (1) snow (not used); (2) soil moisture (computes actual evapotranspiration, soil moisture, and groundwater recharge); (3) response function (calculates runoff and groundwater levels); and (4) routing (calculates the distribution of runoff for a given time series). Model calibration used 2018 data, and validation was performed with data from 2017. Performance criteria included the Nash–Sutcliffe efficiency (NSE) normalized statistic [22] combined with the coefficient of determination ($R^2$). The past hydrological condition of the basin was then simulated using climatic data from the Boromo and Dano stations (Figure 1) for the period 1971–2018 (Figure 4).

### 2.5. Water Availability Evaluation during the Critical Phase of Rice Development

The flowering and grain filling growth stages of rice are the most critical phases of its development. Water stress during these phases can be seriously detrimental to the rice yield [10]. In the Dano region, rice varieties with a growing period of 120 days are the most cultivated. By considering the following 10-day periods of rice sowing seed (21–30 June, 1–10 July, and 11–20 July), the critical periods for rice development ranged from 11–20 September to the 11–20 October. The total discharge of water during these 10-day periods was obtained. Trends in these data collection periods were evaluated using the Mann-Kendall and Sen slope tests. The availability of water in terms of discharge, level, and volume needed for effective rice development in the Lofing inland valley was then assessed.

## 3. Results and Discussion

### 3.1. Farmers' Perceptions of Changes in Climatic Hazards in the Lofing Inland Valley

Farmers have both positive and negative consensual perceptions of climatic hazards. According to them, no severe drought has been observed since 1974, except in 1984, when a famine was experienced. Since 1996, extreme rains able to destroy houses have not been observed, except in 2015, when a long-lasting and heavy rainfall event was recorded. Events perceived negatively were deemed to be of the greatest importance. Farmers consider that the heat waves initially experienced mainly in April have shifted to March and that they last almost the entire year. The weather is therefore perceived as becoming warmer. Furthermore, the farmers perceive there to be changes in annual rainfall patterns because, according to them, rainfall was previously well-distributed throughout the rainy season. Now, they believe that the dry season is longer, and the rainy season is shorter and irregular. Dry spells have become more frequent during crop growth and mainly occur during the rice grain formation stage. According to the interviewed farmers, there is currently a decrease in rainfall amount per event, with more thunderstorm winds, but limited rainfall, in comparison to past observations.

The events best-described by the farmers are the catastrophic years, the increasing heat, and the effect of wetland development over the last two years. The consensual perceptions of the degrading rainy season should be considered with caution, given the vagueness of the compared periods and the intensity of the variations. For example, it is virtually impossible to attribute a precise date to references such as "previously", "formerly", or "around 2000", when farmers stated that there have been more severe droughts in the past than at present. Was there a period of a small series of very regular years that was idealized and would now be "referenced"? Did the farmers identify recent problematic years to more comprehensively judge the past, thereby forgetting about the reality of the variability and the change in the process? An in-depth analysis of several timescales of the long climatological series, including a frequency analysis, is required to answer these questions.

Some perceptions are not consensual, namely, the rainfall onset date in 2017 and the effect of the dike and channel rehabilitation in 2017. In fact, 10 of the 17 people interviewed reported an early

rainfall onset, 3/17 interviewees reported a normal onset, and 4/17 interviewees indicated a late rainfall onset in 2017. The least consensual perceptions are paradoxically related to the climate or the water regime of the year. These perceptions address, on the one hand, the location of the respondent's plot in relation to the newly constructed dike, and on the other hand, the expectations that are relative to a farmer's specific needs and workplan. The perceptions of climate risk also do not have the same levels of concern among individual respondents. Few have seen "no change" to the climate. In terms of the motivation for sowing rice rather than transplanting in 2017, respondents cited the climatic risk. However, the perception of climate risk varies, according to the respondents' gender and the level of development of the lowlands (Table 1). Indeed, excluding the developed lowlands, the climate risk was mentioned by 100% of women as the reason for rice transplanting. This result might be due a lack of knowledge about agricultural rainfall onset identification. Early sowing is adopted by farmers to free themselves from rainfall and hydrological hazards (i.e., uncertainty about the moisture conditions of the lowland region). Despite existing water control structures in the developed lowlands (which are designed to enable transplanting), up to 20% of farmers do not wait for good sowing conditions. Men are more restricted by their other farming operations. Rice is of a lower priority, and where there is a competition of labor against cotton, the lowlands are often abandoned. Indeed, the climate risk is less important to them, as it is shifted toward alternative activities. This local information is extremely valuable as it both identifies the concerns of local farmers and provides new information regarding their perceptions and likely responses as a result. It is nevertheless necessary to compare the local farmers' perceptions with measured data, which is independent of the farmers' gender, knowledge, and situation.

**Table 1.** Reasons for the choice of sowing (rather than transplanting) in 2017, according to gender and situation.

| Reasons | Operational Constraints | Social Organization | Climate Risk | Lack of Know-How |
|---|---|---|---|---|
| Undeveloped lowland by men (%) | 43 | 0 | 43 | 14 |
| Undeveloped lowland by women (%) | 0 | 0 | 100 | 0 |
| Developed lowland by men (%) | 63 | 13 | 25 | 0 |
| Developed lowland by women (%) | 80 | 0 | 20 | 0 |

### 3.2. Analysis of Rainfall over the Last 40 Years at the Regional Scale

The mean rainfall recorded at Dano during 2013–2017 by the Dreyer Foundation (951 mm) is similar to that of the period 1970–2013 at Boromo-Gaoua (962 mm). For that reason, the Boromo-Gaoua rainfall stations, which are the closest to Dano, have been used to provide a detailed understanding of the local climatic pattern. Increased water exceedance events (water available to recharge the reserve and water flows) have been observed in Boromo since 1984 (Figure 5a). There has been no increase in rainfall; however, an increase in water excess implies a change in rainfall regime and/or land use. More rainfall in a shorter time period results in an increase in flood risk and less actual evapotranspiration. In Gaoua, the change was small, but similar, to the change depicted in Boromo, except that the water excess did not increase. Figure 5b shows the climatic balance of the last five years in Dano. There was a high seasonal and interannual variability of rainfall during this period. In 2013, there was no pre-humid period, the water excess was limited, and an early cessation of the season occurred, while in 2014, the pre-humid period occurred in mid-July. In 2015 and 2016, there was no pre-humid period, and the risk of inundation was high. A pre-humid period was detected in early September 2017, with limited water excess.

The period of rainfall uncertainty is longest during the pre-humid season. The hazard zone corresponds to rainfall being less than half of the potential evapotranspiration between the 25 and 75 quantiles of ten days of rainfall. The season profile is asymmetric, meaning that the cessation of the season is more predictable than its onset and that the early rainy season provides more information on

the rainy season duration. An ideal opportunity for an informed choice of season length, mainly if there is the potential to irrigate, is provided based on this data.

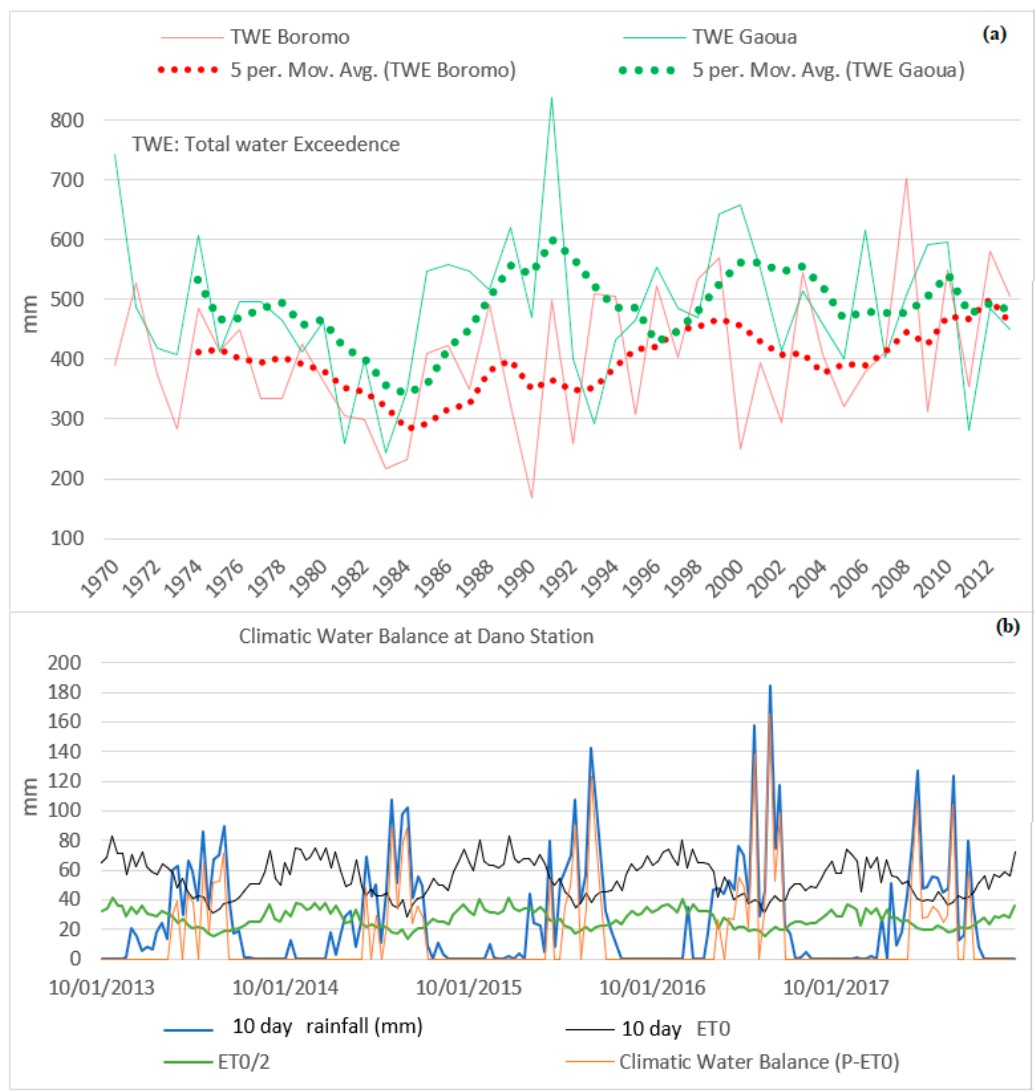

**Figure 5.** Water excess at Boromo and Gaoua rainfall stations from 1970 to 2012 (**a**) and the climatic water balance at Dano station (2013:2017) (**b**).

*3.3. Temperature Analysis at the Regional Scale*

The Sen slopes estimated for each month by considering the minimum and the maximum temperatures at Boromo and Gaoua from 1970 to 2013 indicate a tendency toward a warmer atmosphere. During the rainy season (August to October), the average minimum temperature and the average maximum temperature increased at a rate of 0.031–0.035 and 0.005–0.016 °C yr$^{-1}$, respectively, which is in line with the global observation [23,24]. The minimum temperature increased faster (around two times) than the maximum temperature during the rainy season. Climate change causes increasing air temperatures and evapotranspiration, increases the risk of intense rainstorms, and increases the risk of heat waves associated with drought [25]. An increase in temperature was also observed in the Beninese part of the Niger basin by Badou et al. (2017). An increase in temperature implies a higher level of evapotranspiration, a higher water demand for crops, and a lower level of water exceedance. An increasing temperature will exhibit a larger impact on the grain yield than on vegetative growth and will reduce the ability of the crop to efficiently fill the grain or fruit [26]. This output has the

potential to inform famers and stakeholders in framing appropriate policies for rice intensification in the region. The minimum relative humidity (RH) increases from July to October at Boromo and from August to December at Gaoua. The maximum RH decreases from May to September at Boromo and from July to September at Gaoua.

### 3.4. Potential of Watering/Drainage of the Channels' System in the Lofing Inland Valley

High seasonal and interannual variabilities of discharge were observed at the Lofing-Radier outlet (Figure 6). Throughout the growing season, the river provides enough water to satisfy the requirements of rice (100 L s$^{-1}$ is needed to irrigate 30 ha) (Figure A1). Irrigation should be possible whenever necessary, mainly during the dry spell period. Irrespective of the rice sowing date, the critical period (end of the rice cycle) requiring irrigation varies between the 10th of September and 20th of October. Channel dimensioning is challenging in rice cultivated in inland valleys. There must be a trade-off between the necessities of the discharge peak flow, while maintaining a water level in the channels required for direct irrigation (through, for instance, the use of cofferdams), but also maintaining the wetness of cultivated parcels. The drainage capacity of the channels is large enough for discharging the peak flows arriving from the Dano basin at the outlet of Lofing, while its irrigation potential is problematic. Although the inflow into the channel system was adequate, the water level in the channels is problematic. The minimum water level required in one of the main channels for irrigation is 35 cm. In 2017, the water level in the channels rarely exceeded 35 cm (Figure A2), suggesting that the installation of cofferdams to increase water levels would be beneficial. The implementation of cofferdams, however, may result in additional problems, such as hindering the channel drainage function during peak flow. Frequency analysis has shown that the likelihood of obtaining a high flow during the critical period is low, and mainly occurs after the 21st of September. To ensure that irrigation at the end of the rice cycle is sufficient, different mobile cofferdam types require testing to ensure their efficiency and acceptability in terms of cost and the capacity of farmers to implement the technology.

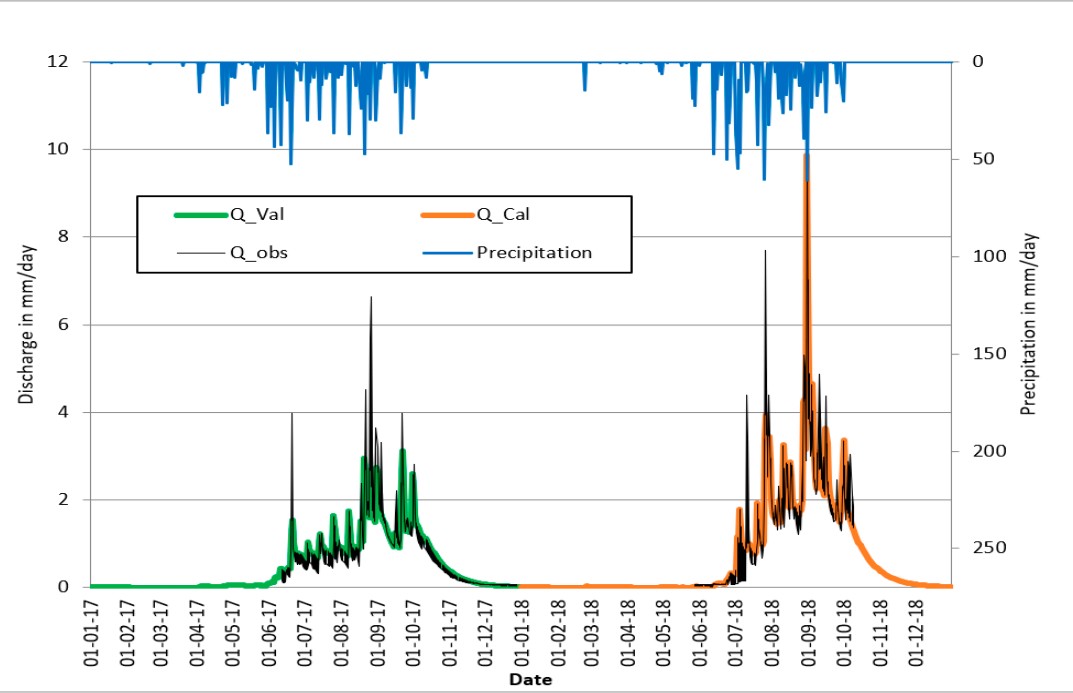

**Figure 6.** Observed and simulated discharges produced by the HBV model (Q_Val, Q_Cal, and Q_obs corresponding to the validation discharge, the calibration discharge, and the observed discharge, respectively).

### 3.5. Model Calibration and Validation

The HBV model was calibrated for the year 2018, and the simulated discharge was compared to the observed discharge using the numerical and visual criteria. The observed and simulated discharges are similar (Figure 6). NSE values for calibrated/validated data were 0.75/0.70, with a coefficient of determination of 0.75/0.73 and a logarithm of NSE of 0.74/0.85. The high values imply a strong performance of the HBV model for the two years of observed discharge. Both high discharge and recession discharge (lower discharge) series were accurately simulated, although it is acknowledged that a greater number of discharge observations are required to increase the accuracy in the future.

### 3.6. Hydrological Condition of the Lofing Upstream River

The calibrated and validated HBV model simulated discharge of the Lofing-Radier River for the period 1971–2018. The water balance components, precipitation, discharge, and actual evapotranspiration are shown in Figure 7. The results of the Mann-Kendall trend test applied to the discharge statistic are shown in Table 2. No statistically significant trend at the 5% level was found in the total annual discharge for the period 1971–2018, indicating that the hydrological regime of the catchment did not vary at the annual scale. Nevertheless, a small annual increasing rate of the discharge of 0.0074 mm per day (2.7 mm yr$^{-1}$) was evident, implying a constant availability of water at the annual scale. The flowering and grain filling of rice seeded between the 20th and 30th of June occurred between the 11th and 20th of September. This stage of rice growing is critical since water stress experienced in this period may drastically reduce the rice yield. No significant increase in the water stress level is found during this period. Between the 21st and 30th of September, the third quartile displays an increase in the risk of excess water levels. In October, there is an increase in water resource availability, which is mainly beneficial for rice production during this critical stage.

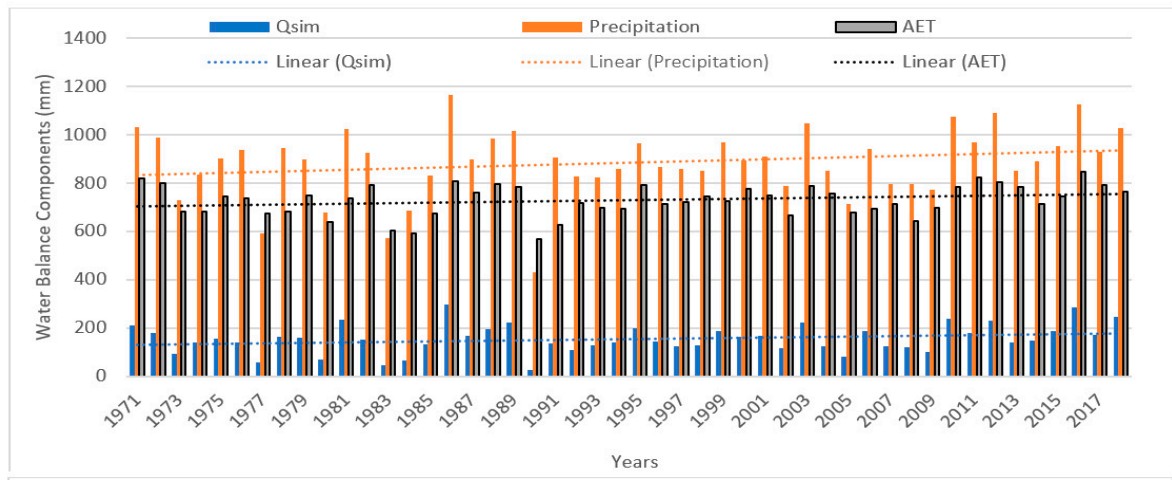

**Figure 7.** Water balance components: precipitation, simulated discharge (Qsim), and actual evapotranspiration (AET) for the period 1971–2018.

**Table 2.** Sen slope (SS) and total 10-day water discharges (see Section 3.3) in different conditions for 1971–2018. * indicates a significant trend obtained through the Mann-Kendall test.

|  | 11–20 September | | 21–30 September | | 1–10 October | | 11–20 October | |
|---|---|---|---|---|---|---|---|---|
| Test implemented (1971–2018) | SS/MK * | 10 day water (mm) | SS/MK * | 10 day water (mm) | SS/MK * | 10 day water (mm) | SS/MK * | 10 day water (mm) |
| 1st Quartile (Dry Condition) | 0.006 | <12.7 | 0.009 | <11.7 | 0.009 * | <9.2 | 0.006 * | <6.4 |
| Median (Normal Condition | 0.009 | 19.5 [12.7, 24.1] | 0.010 | 16.8 [11.7, 20.5] | 0.010 * | 12.6 [9.2, 5.7] | 0.007 * | 9.2 [6.4, 10.4] |
| 3rd Quartile (Wet Condition) | 0.014 | >24.1 | 0.016 * | >20.5 | 0.013 * | >15.7 | 0.008 * | >10.4 |
| Sum | 0.120 | - | 0.125 | - | 0.127 * | - | 0.070 * | - |

The water level in the river at the gauging station decreased from the 10th of September to 10th of October (Table 2). Therefore, if there is enough rainfall, sowing the rice during 10–30 June will be optimal to take advantage of the higher surface water flows that can be mobilized to irrigate and meet the crops' water requirements during the critical phases of flowering and formation-filling of the grains. Lower flow rates can be utilized to irrigate the crop during the critical phases if sown between the 1st and 20th of July.

## 4. Conclusions

Rainfall events exceeding 100 mm and flash flooding are becoming more frequent, leading to severe damage to crop production and water infrastructure. Special attention must therefore be given to the design of water control structures to ensure their flexibility and sustainability in discharging floods while avoiding overdrainage during dry spells. In this study, we analyzed the hydroclimatic conditions of the study area Dano, Burkina Faso, and the implication for rice production in the region. There was no significant increase in annual rainfall for the period of 1970–2013; however, an increasing delay in the onset of the rainy season (with a decreasing pre-humid season duration) was observed. This causes difficulties in predicting the onset due to the high temporal variability of rainfall in the studied region. As a result, a warming trend was observed for the past 40 years, raising questions about its negative impact on very intensive rice cultivation packages. During the rainy season (August to October), the average minimum and maximum temperatures increased by 0.031 and 0.016 $^{\circ}$C yr$^{-1}$, respectively, comparable to global observations. The maximum relative humidity decreased due to this increase in temperature, while the sunshine duration also decreased. Farmers have both positive and negative consensual perceptions of climatic hazards. The HBV hydrological model indicated no significant increase in water discharge; however, the total 10-day water level observed between the 11th of September and 20th of October, corresponding to the critical flowering and grain filling phases of rice growth, showed an increasing trend for the period 1971–2018.

The sowing of rice during the 10–30 June has been identified as optimal in order to benefit from the higher surface water flows, which can be used to irrigate and meet the crop water requirements during the critical phase outlined. The installation of cofferdams to increase water levels would be beneficial, subject to them not hindering channel drainage during peak flow, although water flow after the 21st of September was generally insufficient to be deemed an issue. To ensure that irrigation at the end of the rice cycle is sufficient, different mobile cofferdam types require testing to ensure their efficiency and acceptability in terms of cost and the capacity of farmers to implement the technology. The results of this study will be useful to rural communities, as well as decision makers, in framing agricultural risk management in the study region and devising policy for rice intensification in lowland areas. Further data collection is required to improve the HBV model output and to account for climate and land change effects on rice production in Dano, Burkina Faso.

**Author Contributions:** Conceptualization, A.Y.B. and G.S.; methodology, A.Y.B., G.S., J.H., and B.D.; formal analysis, A.Y.B., J.H., Y.Y., G.S., and B.D.; writing—original draft preparation, review, and editing, A.Y.B., J.H., G.S., Y.Y., B.L., J.L.F., L.O.S., J.E.T., and B.D. All authors have read and agreed to the published version of the manuscript.

**Funding:** This research received no external funding.

**Acknowledgments:** The authors are grateful for the financial support provided by the French Agency for Development (AFD) under the auspices of the AGRICORA initiative and GENERIA project. They thank the German Federal Ministry of Education and Research (BMBF) for supporting the WASCAL program.

**Conflicts of Interest:** The authors declare no conflicts of interest.

## Appendix A

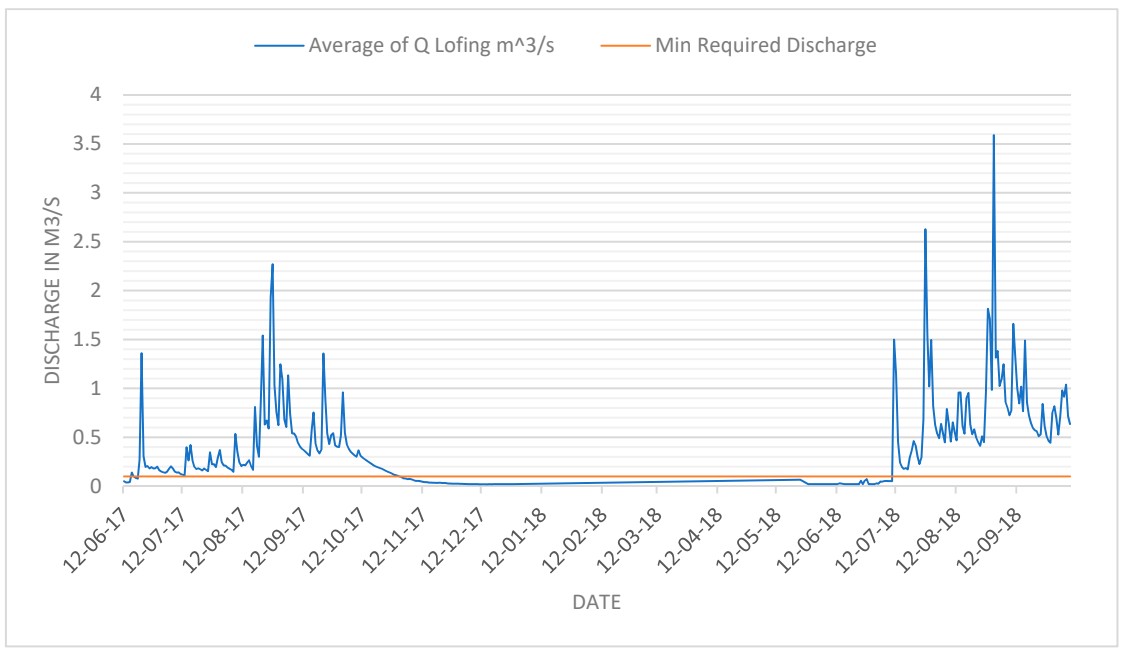

**Figure A1.** Comparison of the available discharge and irrigation water requirement for the 30 ha Lofing inland-valley.

## Appendix B

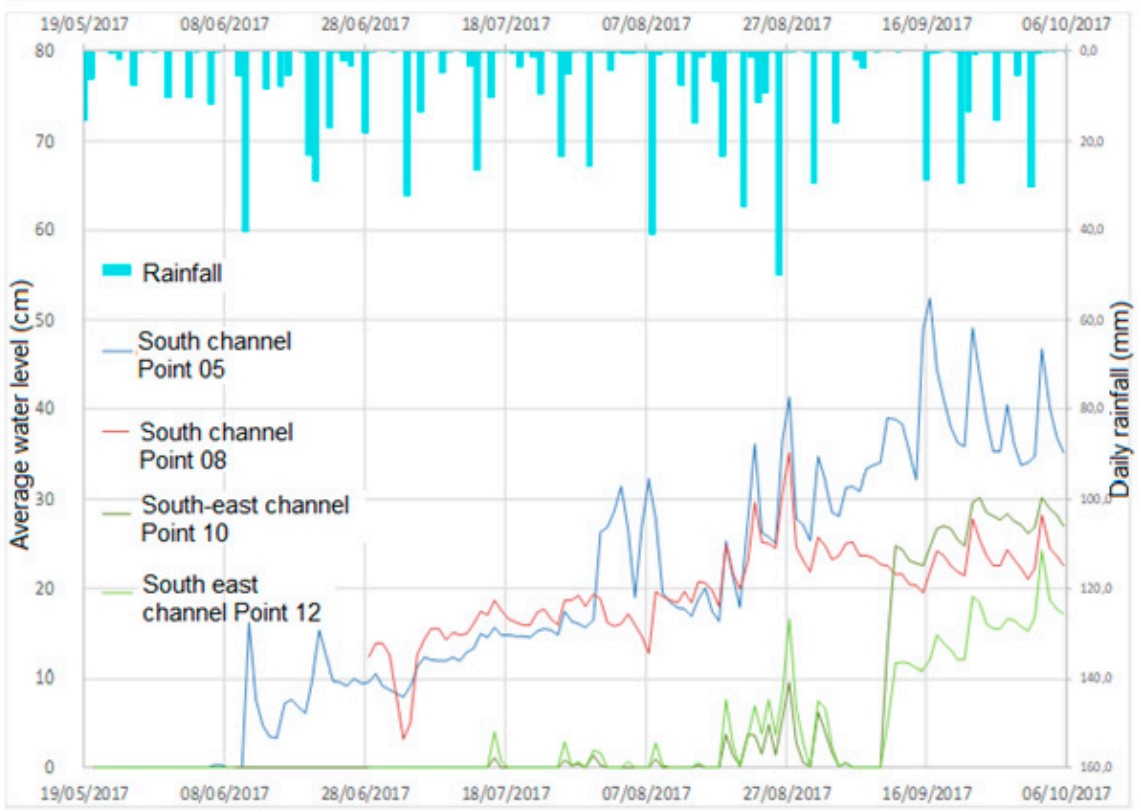

**Figure A2.** Comparison of the observed and required water height in the irrigation/drainage channels of Lofing inland-valley. Points represent the water level measurement locations.

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
