# Peer review of "Managing New Risks of and Opportunities for the Agricultural Development of West-African Floodplains: Hydroclimatic Conditions and Implications for Rice Production"

_climate, doi:10.3390/cli8010011_

Round 1

Reviewer 1 Report

I cant see the previous comments and improved article with track changes, although I tried many time to open the link provided. As, it is quite long when I previously reviewed it and I can not remember my all suggestions. Although current version looks improved and short as well. It looks authors reduced it little bit more as suggested. All section looks improved but still there are few specific comments and suggestions are provided in attached reviewed articles for its improvement.

Author Response

Point 1: I can’t see the previous comments and improved article with track changes, although I tried many time to open the link provided. As, it is quite long when I previously reviewed it and I can not remember my all suggestions. Although current version looks improved and short as well. It looks authors reduced it little bit more as suggested. All section looks improved but still there are few specific comments and suggestions are provided in attached reviewed articles for its improvement.

Reply 1: We thank the reviewer for this comment. We are sorry for the fact that the reviewer was not able to access the track change version of the article. Nevertheless, the submitted version of the manuscript was substantially improved as the reviewer acknowledged.  

Point 2: Tittle may little bit modified as study is focused on rice crop not all agricultural development.

Reply 2: We thank the reviewer for this comment. The authors agree with the reviewer that the study focused on rice crop and this justify the second portion of the manuscript title were rice crop was clearly mentioned.

Point 3: Line 38, replace "optimal" word with another suitable word.

Reply 3: We thank the reviewer for this comment. The word “optimal” describe well our thoughts.

Point 4: Line 39: critical flowering and grain filling phases, please also mention the days here according to crop.

Reply 4: the requested information is provided in section 2.5. Water availability evaluation during the critical phase of rice development.

Point 5: check space between these two sentences

Reply 5: The additional spaces between two sentences were deleted throughout the manuscript.

Point 6: Line 111: is it written properly, please check.

Reply 6: The authors confirm that ET0 is the right symbol.

Point 7: Line 121: again, picture quality is not good.

Reply 7: The picture quality was improved.

Point 8: Line 185: is 17 farmers represents the complete study region.

Reply 8:  We do agree with the reviewer that the data sample size is not suffisante for a significant statistical analysis. This survey helps to get at least the perception of the famers on the climate variability and change. These perceptions were further compared with the scientifically based results.

Reviewer 2 Report

This revised work is significantly improved. Most of the comments are incorporated. However; the part with the survey remains problematic. This very low number of farmers is leads to results which are biased and not significant. This part has no added value for the rest of the text and must be removed.

Author Response

This revised work is significantly improved. Most of the comments are incorporated. However; the part with the survey remains problematic. This very low number of farmers is leads to results which are biased and not significant. This part has no added value for the rest of the text and must be removed.

Reply 1:  We thank the reviewer for this comment. We do agree with the reviewer that the data sample size is not suffisante for a significant statistical analysis, but this raison is not enough for removing the section on the survey of this study as suggested by the reviewer. This survey helps to get at least the perception of the famers on the climate variability and change. These perceptions were further compared with the scientifically based results.

Reviewer 3 Report

1. Section 2 (Materials and Methods) requires two/three sentences to start the section.

2. Section 2 needs to be rearranged as  2.1 Study Area, 2.1 Data ……2.6 Water availability evaluation during the critical phase of rice development.

3. Text related to the “Figure 3: Methodological framework of the study” in section 2.3 should be moved to right after section 2

Author Response

Point 1: Section 2 (Materials and Methods) requires two/three sentences to start the section.

Reply 1: We thank the reviewer for this comment. The Section 2 (Materials and Methods) was introduced with the following sentence: “This section is divided into five sub-section which are the study area and data used, the various modalities of lowland development: traditional vs modern development, the climate-related local knowledge and hydroclimatic variables’ analysis, the rainfall - runoff modeling and frequency analysis, and the water availability evaluation during the critical phase of rice development”.

Point 2: Section 2 needs to be rearranged as  2.1 Study Area, 2.1 Data ……2.6 Water availability evaluation during the critical phase of rice development.

Reply 2:  The description of the data used in this study is not enough to be considered as sub-section. It was therefore merge with the study area to have the current “Study area and data used” sub-section

Point 3: Text related to the “Figure 3: Methodological framework of the study” in section 2.3 should be moved to right after section 2

Reply 3: The figure 3 was the first time cited in the section 2.3. This justify it position at the end of section 2.3. “Climate-related local knowledge and hydroclimatic variables’ analysis”.

This manuscript is a resubmission of an earlier submission. The following is a list of the peer review reports and author responses from that submission.

Round 1

Reviewer 1 Report

Reviewer’s Comments:

An interesting article, however, it requires major changes to be published. I am recommending it for publication with major revision as I think the article is important for water management and adaptation. 

Major Comments:

In the title “African flood plains” should be replaced by “West African flood plains.”

Introduction should be rearranged to establish a clear link among African floodplains, Hydroclimatic risks, agriculture (rice) of Burkina Faso’s and research objectives.

A brief description of the Figure 1 is required in the text.

Section 2.1: “Study area and Data used” can be separated as two sections: “Study area” and “Data used.”

Data source should be mentioned in the text.

Incorporation a graph of rainfall and temperature patter in the paper will help the reader to understand the climate of the study area.

There is Picture1. Picture 2 should be picture1. Picture 1 should be arranged clockwise, like Picture1(a), Picture 1(b), Picture 1(c), and Picture 1(d). If possible, present pictures should be replaced by the high-resolution pictures.

A methodological flowchart should be used to visualize the complex methodological procedures of the study.

The result and discussion part should be separated to smooth reading. In present form it is very complex as a reader to keep track of the result and discussion.

In discussion and conclusion, the implication of this study should be captured very clearly with important points.

Minor Comments

Figure 1 should be rearranged by Figure a, b, c, d in clockwise. If possible, the legend should be inside the frame of the map. 

English of the paper should be checked, and the presentation of the paper requires a complete attention of the authors.  

Reviewer 2 Report

This works studies the impact of hydroclimatic risks in a region of Burkina Faso and its implications for rice production. The idea is very good, however this paper suffers from many major shortcomings.

The title is not really adequate, as this work is not really studies the hydroclimatic risks, but more the hydroclimatic conditions. The part with the survey must be completely removed. The questionnaire is not provided. The opinion of only 17 farmers is not representative and thus not significant. There is no explanation on the collection method. There is an extensive literature on how people recall past events. I suggest to the authors to better read this literature. The results given here are expected (and not significant). There is no comparison of the memory on past events and the real meteorological conditions of those years. There is no statistical treatment of the results (descriptive statistics, etc). The results are severely biased due to the small sample. The calibration of the model with only one year is problematic. Please give the uncertainties of the results of using only one year. A time series presentation of the rainfall and temperature must be provided. Rainfall (3.2) Please explain why these time periods are used? A analysis of the rainfall time series must lead to the periods to be used. This analysis is missing. The fluctuations of the curves in figure 3.2 are quite significant and the conclusions given here are not supported from these figures. Temperature (3.3) A time series analysis is missing. There is no argumentation on the time periods used. Why only the increases in temperature are shown? As the values of figures 5 and 6 statistically significant? Even if they are, the overall concept is not sound (arbirtary time periods, no projections in the future, etc.) The analysis is provided to provide management of risks but: There is no projection in the future. There is no policy implications and no real suggestions.

Minor comments

Average annual temperature from 21 to 38C? Please check.

Please check figures 1b and c.

Please indicate the location of Boromo.

Temperature is monitored since 1922 in Burkina Faso? Please check.

Please check figure 3.

Please check funding and acknowledgements.

Reviewer 3 Report

Article "Managing new risks and opportunities of agricultural development of African floodplains: Hydro-climatic risks and implications for rice production" is a good article, written in comprehensive ways. Although it is a good article but lacking novelty, and scientific soundness. Although many data set are complied in it but still it is lacking novelty as it is a local scale study.
Introduction section is too general, need to specify the things and mainly focus on studied parameters like climate, environment, water, hydro-climate, rice crop and production aspects need to mention.
Materials and Methods
Many confusing things, there is need to explain many aspects especially related to data, data type and duration.
Specific comments can be seen in the relevant section please.

Result are too general, need to specify and reduce it.
Discussion, totally lacking, need to add proper discussion and citation of previous studies.
Detailed and specific comments can be seen in attached reviewed file.

Reviewer 4 Report

The paper “Managing new risks and opportunities of agricultural development of African floodplains: Hydroclimatic risks and implications for rice production” is interesting.

The paper assesses the hydroclimatic risks and implications for floodplain rice production in southwestern Burkina Faso to make informed decisions regarding floodplain development and to participate in the co-construction of new climate adaptation options. Statistical methods combined with rainfall runoff modeling were used to analyze the hydroclimatic conditions of the study area. Moreover, spatial and temporal water availability for crop growth is assessed for an innovative and participatory water management technique. From 1970-2013, an increasing delay of the onset of the rainy season (with a decreasing pre-humid season duration) occurred, causing difficulties in predicting the onset due to the high temporal variability of rainfall in the studied region. A warming trend is observed over the past 40 years, raising questions about its negative impact on very intensive rice cultivation packages. The analysis of the hydrological condition of the basin through the successfully calibrated and validated hydrological model HBV indicated no significant increase in water discharge. Rainfall events of more than 100 mm and flash flood events are becoming more frequent, leading to high damages to crop production and water infrastructures. Special attention must therefore be given to the design of water control structures to ensure their flexibility and sustainability in discharging floods while avoiding overdrainage during dry spells.

     Some minor comments

The extensive editing of English language and style is required.. If the author builds a model to analyze, maybe it's better. For example, as the following literatures.

Mojie Li, Zhifu Mi, D'Maris Coffman, Yi-Ming Wei, Assessing the policy impacts on non-ferrous metals industry's CO2 reduction: Evidence from China, Journal of Cleaner Production, Volume 192, 10 August 2018, Pages 252-261

Zeng S, Nan, X., Liu, C., Chen J., The response of the Beijing carbon emissions allowance price (BJC) to macroeconomic and energy price indices, Energy Policy, 2017,106: 111-121.

    Mingxing Sun, Yutao Wang, Lei Shi, Jiří Jaromír Klemeš. Uncovering energy use, carbon emissions and environmental burdens of pulp and paper industry: A systematic review and meta-analysis. Renewable and Sustainable Energy Reviews, 2018 (92): 823-833.

    Zeng S, Jiang, C., Ma, C., Su, B., Investment Efficiency of the New Energy Industry in China, Energy Economics, 2018, 70:536-544.

I think It is valuable after revise for accepted publication.

Author Response

Dear reviewer,

many thanks for your helpful comments on our manuscript. We were able to incorporate all your suggestions during the review. Please, find below our detailed answer to your comments. Your suggestions and remarks have really improved our paper. We hope that you find your comments sufficiently considered and we would like to thank you for your support.

Aymar Yaovi Bossa, Jean Hounkpè, Yacouba Yira, Georges Serpantié, Bruno Lidon, Jean Louis Fusillier, Luc Olivier Sintondji, Jérôme Tondoh, Bernd Diekkrüger

Comments and Suggestions for Authors

Point 1: The paper “Managing new risks and opportunities of agricultural development of African floodplains: Hydroclimatic risks and implications for rice production” is interesting.

R1: We thank the reviewer for this comment

The paper assesses the hydroclimatic risks and implications for floodplain rice production in southwestern Burkina Faso to make informed decisions regarding floodplain development and to participate in the co-construction of new climate adaptation options. Statistical methods combined with rainfall runoff modeling were used to analyze the hydroclimatic conditions of the study area. Moreover, spatial and temporal water availability for crop growth is assessed for an innovative and participatory water management technique. From 1970-2013, an increasing delay of the onset of the rainy season (with a decreasing pre-humid season duration) occurred, causing difficulties in predicting the onset due to the high temporal variability of rainfall in the studied region. A warming trend is observed over the past 40 years, raising questions about its negative impact on very intensive rice cultivation packages. The analysis of the hydrological condition of the basin through the successfully calibrated and validated hydrological model HBV indicated no significant increase in water discharge. Rainfall events of more than 100 mm and flash flood events are becoming more frequent, leading to high damages to crop production and water infrastructures. Special attention must therefore be given to the design of water control structures to ensure their flexibility and sustainability in discharging floods while avoiding overdrainage during dry spells.

     Some minor comments

Point 2: The extensive editing of English language and style is required.

R2: We thank the reviewer for this comment. The English of the manuscript was improved. The manuscript was initially edited for English language by AJE (see attached the certificate)

Point 3: If the author builds a model to analyze, maybe it's better. For example, as the following literatures.

Mojie Li, Zhifu Mi, D'Maris Coffman, Yi-Ming Wei, Assessing the policy impacts on non-ferrous metals industry's CO2 reduction: Evidence from China, Journal of Cleaner Production, Volume 192, 10 August 2018, Pages 252-261

Zeng S, Nan, X., Liu, C., Chen J., The response of the Beijing carbon emissions allowance price (BJC) to macroeconomic and energy price indices, Energy Policy, 2017,106: 111-121.

    Mingxing Sun, Yutao Wang, Lei Shi, Jiří Jaromír Klemeš. Uncovering energy use, carbon emissions and environmental burdens of pulp and paper industry: A systematic review and meta-analysis. Renewable and Sustainable Energy Reviews, 2018 (92): 823-833.

    Zeng S, Jiang, C., Ma, C., Su, B., Investment Efficiency of the New Energy Industry in China, Energy Economics, 2018, 70:536-544.

I think It is valuable after revise for accepted publication.

R3: We thank the reviewer for this comment. The provided literatures by the reviewer were considered to improve the manuscript.

Round 2

Reviewer 1 Report

The paper needs more editing to give a better shape and also the authors need to establish a clear flow from one section to another section of the paper so that reader can read the paper with ease.

Author Response

Dear reviewer,

many thanks for your helpful comments on our manuscript. We were able to incorporate all your suggestions during the review. Please, find below our detailed answer to your comments. Your suggestions and remarks have really improved our paper. We hope that you find your comments sufficiently considered and we would like to thank you for your support.

Aymar Yaovi Bossa, Jean Hounkpè, Yacouba Yira, Georges Serpantié, Bruno Lidon, Jean Louis Fusillier, Luc Olivier Sintondji, Jérôme Tondoh, Bernd Diekkrüger

Comments and Suggestions for Authors

Point: The paper needs more editing to give a better shape and also the authors need to establish a clear flow from one section to another section of the paper so that reader can read the paper with ease.

R1: We thank the reviewer for this comment. The English language of the manuscript as well as its content were improved. The manuscript was initially edited for English language by AJE (see attached the certificate)

Reviewer 2 Report

The authors replied to my comments, but I would expect a better quality of the answers. This paper can be published now, but I would like to suggest to the authors to try to have a higher implication to clarify this paper (also for future works).

Please check English language using a native speaker; some phrases are not so clear.  

Author Response

Dear reviewer,

many thanks for your helpful comments on our manuscript. We were able to incorporate all your suggestions during the review. Please, find below our detailed answer to your comments. Your suggestions and remarks have really improved our paper. We hope that you find your comments sufficiently considered and we would like to thank you for your support.

Aymar Yaovi Bossa, Jean Hounkpè, Yacouba Yira, Georges Serpantié, Bruno Lidon, Jean Louis Fusillier, Luc Olivier Sintondji, Jérôme E. Tondoh, Bernd Diekkrüger

Comments and Suggestions for Authors

The authors replied to my comments, but I would expect a better quality of the answers. This paper can be published now, but I would like to suggest to the authors to try to have a higher implication to clarify this paper (also for future works).

Point 1: Please check English language using a native speaker; some phrases are not so clear. 

R 1: We thank the reviewer for this comment. The English language of the manuscript as well as its content were improved. The manuscript was initially edited for English language by AJE (see attached the certificate)

Reviewer 3 Report

I am sorry to say my suggestions and comments have not been addressed. Please look inside attached reviewed report 1.

Author Response

Dear reviewer,

many thanks for your helpful comments on our manuscript. We were able to incorporate all your suggestions during the review. Please, be informed that the file in which your comments were inserted was not send to us in the review round 1. This justifies the fact that some of your comments were not previously considered. Please, find below our detailed answer to your comments. Your suggestions and remarks have really improved our paper. We hope that you find your comments sufficiently considered and we would like to thank you for your support.

Aymar Yaovi Bossa, Jean Hounkpè, Yacouba Yira, Georges Serpantié, Bruno Lidon, Jean Louis Fusillier, Luc Olivier Sintondji, Jérôme Tondoh, Bernd Diekkrüger

Point 1: Article "Managing new risks and opportunities of agricultural development of African floodplains: Hydro-climatic risks and implications for rice production" is a good article, written in comprehensive ways.

R1: We thank the reviewer for this comment. Well noted

Point 2: Although it is a good article but lacking novelty, and scientific soundness. Although many data set are complied in it but still it is lacking novelty as it is a local scale study.

R2: We thank the reviewer for this comment. We do not agree with the justification provided by the reviewer to indicate that the manuscript lack novelty and scientific soundness. In applied science such as hydrology, a study area will always be necessary. A local scale study does not necessary lack novelty. In fact, this study is among the first of its kind to be done in the south west region of Burkina Faso. The findings of this study will be very useful for local farmers and decision makers in intensifying rice production in the South West region of Burkina Faso and beyond.    

Point 3: Introduction section is too general, need to specify the things and mainly focus on studied parameters like climate, environment, water, hydroclimate, rice crop and production aspects need to mention.

R3: The introduction was rewritten to consider the reviewer observation: see for instance the highlighted parts (lines: 39 – 47; 66 – 72).  

Point 4: Materials and Methods

Many confusing things, there is need to explain many aspects especially related to data, data type and duration. Specific comments can be seen in the relevant section please.

R4: We thank the reviewer for this comment. The information related to the data (types, sources and duration) are clearly indicated in the Material and Method section. See the highlighted section: lines 127 – 133.  

Point 5: Result are too general, need to specify and reduce it. 

R5: We thank the reviewer for this comment. We do not agree with this comment as the reviewer did not provide any justification on how the results are general.

Point 6: Discussion, totally lacking, need to add proper discussion and citation of previous studies. 
Detailed and specific comments can be seen in attached reviewed file.

R6: We thank the reviewer for this comment. The discussion was improved throughout the manuscript.

Specific comments can be seen in the relevant section please in the manuscript.

Point 7: Line 20. Please replace co-construction with other suitable word like development or any other

R 7: The word ‘co-construction’ was replaced by word ‘development’

Point 8: Line 20. Please mention the name of statistical models ?

R 8: The Statistical methods were mentioned: Mann Kendall test, Sen slope estimator, Frequency analysis

Point 9: Line 21. Same as above comment, which models ?

R 9: The model used was mentioned: HBV

Point 10: Line 23. innovative and participatory water management technique: Which techniques?

R 10: We thank the reviewer for this comment. Please, refer to the rest of the manuscript since everything cannot be detailed in the abstract.

Point 11: Line 32. What about results of the study? What about future recommendation?

R 11: We thank the reviewer for this comment. Lines 25 – 32 provide some results of the study while lines 32 – 34 indicate recommendation. Other recommendations are provided in the conclusion section. 

Point 12: Line 36. Introduction.

It is too general, please specify it and mainly focus on studied parameters like climate, environment, water, hydro-climate, rice crop and production aspects need to mention.

R 12: We thank the reviewer for this comment. The introduction was rewritten to consider the reviewer observation: see for instance the highlighted parts (lines: 39 – 47; 66 – 72)

Point 13: Line 79, Materials and Methods

Line 79. Many confusing things, there is need to explain many aspects especially related to data, data type and duration. Specific comments can be seen below

R 13: We thank the reviewer for this comment. The information related to the data (types, sources and duration) are clearly indicated in the Material and Method section. See the highlighted section: lines 125 – 130.

Point 14: Line 100-104, Is it necessary to mention all this significance of rice

R 14: We thank the reviewer for this comment. We think these information help to better understand the study area

Point 15: Line 108, Data of different variables are for different time periods like few parameters data are from 1922 to 2016 while few parameters data are 2017-18 only?

R 15: We thank the reviewer for this comment. Statistical analysis was done using the long-term climate variables for the period 1922 – 2016. The calibration and the validation of the HBV model was done based on the data of 2017-2018 (discharge and climate variables) which were measured during the project implementation. This justifies the difference in data length.

Point 16: Line 128. Please follow the same format as Figure not use the word picture

R 16: We thank the reviewer for this comment. Figure format has been changed following the reviewer’ s comment. The numbers on the figure have been replaced by the letters.

Point 17: Lines 144- 155; Model description; It is general details but lacking scientific soundness. Need to specify the things.

R 17: We thank the reviewer for this comment. The sub-section 2.2. describes the various modalities of lowland development, accounting for the traditional and modern lowland development to help to reader better understand the context of the study from different point of views. Therefore, it came just after the description of the study area.

Point 18: Line 179. Model description is required to mention as well.

R 18: We thank the reviewer for this comment. The model was more described to account for the reviewer comment. See lines 208 – 212.

Point 19: Line 186. Which conditions?

R 19: We thank the reviewer for this comment. Hydrological condition of the basin refers to its hydrologic behavior which was developed in the section 3.6 (Hydrological condition of the Lofing upstream river).

Point 20: Lines 196-199 Confusing

R 20: We thank the reviewer for this comment. This sentence was reformulated, see lines: 224 – 229.

Point 21: Line 225, Farmer perceptions is okay but i think authors should analyze the past climate data then picture will be very clear. Farmer generally reply such type of answers.

R 21: We thank the reviewer for this comment. Past climate data were also analyzed and compared to farmer perceptions in the results section

Point 22: Line 294. Figure 2; Picture resolution is low, further decade should be mention on X-axis instead of number

R 22: We thank the reviewer for this comment. Figure with better resolution have been provided. To avoid overburden the figure, the use of decade numbers on the X-axis was preferred.  There are 36 decades in a year; providing the dates for each decade would make the figure unreadable.

Point 23: Line 311. Fig. 3 X-axis of TWE graph is not readable, need to clear it. Line 318-320, Please specify the decades instead of writing the numbers

R 23: We thank the reviewer for this comment. The figure 3 X-axis was provided in the right format. See the new figure 5. There are 36 decades in a year; providing the dates for each decade would highly increase

Point 24: Line 323. Fig. 4.  Writing of "Hazard Zone; ......" is not clear. Please add legends of decades in the figure for better understanding.

R 24: We thank the reviewer for this comment. The hazard zone here corresponds to the figure portion where the rainfall is less than half of the potential evapotranspiration between the 25 and the 75 quantiles of ten days rainfall.

To avoid overburden the figure, the use of decade numbers on the X-axis was preferred.  There are 36 decades in a year; providing the dates for each decade would make the figure unreadable.

Point 25: Line 348. Same as above comments, need to mention the legends in the graphs

R 25: We thank the reviewer for this comment. The legend was added for each set of figure but not replicated for each of the figures to avoid overloading.

Point 26: Line 402-406; Need to mention the result not only generally mention lower or higher?

R 26: We thank the reviewer for this comment. Lower discharge refers here to the discharge recession period and higher discharge refers to the period were the discharge value are high.

Point 27: Line 408. Fig. 8. What about statistically values after comparison of observed and simulated values

R 27: We thank the reviewer for this comment. The statistical values provided about the calibration and validation of the model are the performance criteria: see lines 398 – 401.

Point 28: Line 464. Please mention the results instead of general writing

R 28: We thank the reviewer for this comment. “A decrease in the maximum relative humidity was found because of the increase in temperature”: this sentence is just relating two different quantities and the rate of increase in the temperature was indicated in the same abstract.

Point 29: Line 472. How much profit?

R 29: We thank the reviewer for this comment. This is not in term of monetary but in term of advantage for the rice

Point 30: Line 475; How much lower?

R 30: We thank the reviewer for this comment. The flow rate was previously quantified, see table 2 for details.
